# Peer Support at the Intersection of Disability and Opioid (Mis)Use: Key Stakeholders Provide Essential Considerations

**DOI:** 10.3390/ijerph19159664

**Published:** 2022-08-05

**Authors:** Joanne Nicholson, Anne Valentine, Emily Ledingham, Sharon Reif

**Affiliations:** 1Institute for Behavioral Health, The Heller School for Social Policy and Management, Brandeis University, Waltham, MA 02453, USA; 2Lurie Institute for Disability Policy, The Heller School for Social Policy and Management, Brandeis University, Waltham, MA 02453, USA

**Keywords:** peer-led, interventions, substance use disorder, opioid use disorder, disability, well-being, intersectionality

## Abstract

Individuals with disabilities may experience higher rates of opioid/substance use disorders (OUD/SUD) than other individuals and are likely vulnerable to unmet treatment needs. Peer support may be beneficial to these individuals, given the evidence of benefits in target populations with similar needs and the potential for overcoming barriers to treatment suggested in the available literature. The objective of this exploratory study was to specify essential considerations in adapting peer support for this population. Diverse key stakeholders (*n* = 16) were interviewed to explore the experiences, needs, and available supports for individuals with disabilities and OUD/SUD. A Peer Support Work Group including members with lived experience advised each component of the study. Semi-structured interview data were content analyzed and memos generated to summarize themes related to the research question. Participants reported extensive professional and personal experience in human services, disability, and recovery. Emergent themes included the importance of accessibility and model fit, the notion of “peerness” and peer match, and essential aspects of peer recruitment, training, and support. An accessible, acceptable, effective model of peer support requires particular attention to the needs of this diverse and varied population, and the contexts in which they are identified, referred, and engaged in services.

## 1. Introduction

Population-based data examining the intersection of disability and opioid (OUD)/substance use disorders (SUD) in the United States (US) is limited, but a growing body of research suggests that individuals with physical, cognitive, and/or sensory disabilities may experience higher rates of opioid/substance use disorders than individuals without disabilities [1,2,3,4,5,6,7,8]. Peer support has emerged as a promising model of service delivery in behavioral health with a growing body of literature suggesting positive outcomes for both the person served and the peer [9,10,11,12,13,14,15,16,17]. A peer support approach has the potential to address specific issues for people with disabilities and OUD/SUD, including barriers to treatment, stigma, and the need for accommodations that are not always recognized or provided by treatment or service providers and programs. It is important to consider the characteristics and needs of individuals living with disabilities and OUD/SUD in making recommendations for peer support models, adaptations, and accommodations.

The potential for opioid misuse and opioid use disorder (OUD) more broadly among individuals with disabilities has garnered increasing attention, considering the current opioid epidemic, particularly given the significant association between prescription opioid misuse and the treatment of chronic non-cancer pain [18]. Individuals with disabilities are a heterogeneous group; some may be significantly more likely to live with chronic pain and to experience co-occurring mental health conditions than their non-disabled peers [3,8], two factors associated with a heightened risk for prescription opioid use and misuse [4]. The COVID-19 pandemic has conveyed significant day-to-day stress, has contributed to challenges in accessing treatment, and has been accompanied by a significant increase in opioid overdose [19,20,21,22].

### 1.1. Barriers to OUD/SUD Treatment for Individuals with Disabilities

It is likely, given the significant gaps between OUD treatment need and receipt of treatment in the US in general [23,24], that unmet need for OUD treatment is also high among individuals with disabilities. Barriers to effective, evidenced-based substance use treatment are commonly experienced among individuals seeking OUD/SUD treatment [23]. These barriers may be significantly greater among individuals with disabilities and may include inaccessible treatment facilities and materials [25,26], and treatment facility staff with little or no training in how best to accommodate the needs of individuals who may have a disability [1,6,27,28]. The stigma associated with disability, in addition to that experienced by people with co-occurring behavioral health conditions, may be related, not only to reluctance on the part of individuals with disabilities to engage in treatment, but also to the ways in which they are perceived by treatment or service providers [6,26].

### 1.2. Peer Support for Individuals with Behavioral Health Disorders

Peer support is offered by a person with lived experience of a behavioral health or medical condition who has navigated the service system and is in recovery. Peer specialists or recovery coaches provide instrumental support (e.g., facilitated access to housing or employment), emotional support (e.g., empathy and concern), informational support (e.g., information and referrals to community resources), and support through affiliations (e.g., connections to recovery community supports, activities, and events) [29,30]. Peer specialists or recovery coaches may follow a structured curriculum; peer support models may include recovery-management check-ups [31] and skills training to improve functioning for individuals with a mental health condition and co-occurring SUD [32]. Alternatively, peer supports may be more informal, focusing on exploring feelings, sharing experiences and challenges, and overcoming social isolation [12]. Peer support may be provided individually or in a group setting, in-person, via e-mail, by telephone, text message, instant messaging, or online [10,12,33].

### 1.3. Peer Support for Individuals with Disabilities

Peer supports have been suggested as beneficial to individuals disabled by spinal cord injuries and traumatic brain injury [34,35], though few published studies exist. To date, most peer support for individuals with spinal cord injuries has focused on increasing physical activity and promoting self-management [34,36,37,38]. Physical wellbeing may be targeted through injury education with the goal of preventing adverse medical sequelae, improving motivation, and providing intellectual stimulation [39]. Literature regarding the potential benefit of peer support for individuals with other disabilities is sparse, suggesting the need for the exploration and elaboration of relevant model components that can be tailored to enhance outcomes for this target population.

### 1.4. Research Question

Given the limited evidence or models for peer support at this intersection, informing adaptation of behavioral health models of peer support requires consultation with diverse stakeholders regarding the unique needs and challenges of the proposed new target population and service context. The objective of this study was to explore and elaborate essential considerations in adapting peer supports for individuals at the intersection of disability and opioid (mis)use through interviews with diverse stakeholders well-acquainted with peer support models in behavioral health and the needs of individuals with disabilities. Our larger goal is to inform the specification of peer supports tailored to this population, and to identify potential barriers and facilitators to implementing such supports.

## 2. Materials and Methods

This study employed an exploratory, developmental design and qualitative methods, appropriate when the research question reflects the goal of developing or adapting an intervention or practice approach to new conditions or target populations [40]. Diverse stakeholders were interviewed to elaborate the experiences, needs, and peer supports available to individuals with disabilities and opioid (mis)use. A Peer Support Work Group, including members with lived experience of disability and OUD/SUD, advocates, policy makers and program providers, advised the research team during each component of the study, providing suggestions, feedback and recommendations regarding next steps, and the assisting with the interpretation of findings. The study protocol was reviewed and approved by the university Institutional Review Board. Informed consent was obtained verbally from stakeholders prior to their participation in the study, with opportunity provided to address any questions or concerns. Participants were offered a USD 50 gift card of their choosing for their time and effort.

### 2.1. The Sample and Recruitment

A purposeful sample (*n* = 16) was recruited in 2019 and 2020 (prior to the COVID-19 pandemic) using a snowball approach, to reflect domains and experiences of interest: opioid/substance use and the disabling conditions of mental health disorders, traumatic brain injury, spinal cord injury, Autism, and others. Participants included those with lived experience as a disabled person, advocate, family member, or provider with in-depth personal and/or professional knowledge about the intersection of opioid (mis)use and disability. They had personal and/or professional experience in human services, as service providers, peer specialists, or service recipients. Given the lack of research on this target population, a purposeful sample was considered appropriate. The Peer Support Work Group provided initial recommendations for potential participants, based on their extensive experience and well-developed networks of colleagues. Additional participant recommendations were sought from interviewees. An email invitation was sent to nominees, along with an information sheet describing the study. The name of the person referring the nominee to the study was provided, with permission. Once the person agreed to participate, a follow-up email was sent with a copy of the consent form and background survey. The interview was scheduled and the participant’s preference for mode of interview (e.g., telephone, video conference) was solicited.

### 2.2. Interview Procedures

Interviews were conducted by three trained research team members, experienced in interviewing and qualitative methods. Interviews lasted from 1 to 1.5 h. Accommodations were made as appropriate to facilitate participation. For example, an Autistic participant requested the opportunity to review and respond to interview prompts via text and email. At the start of the interview, the interviewer reviewed the consent form with the participant, responded to any questions, and obtained verbal consent. In addition, participants were asked their permission to audio record the interviews. All participants consented and agreed to audio recording.

A brief background survey was completed with the participant to obtain data regarding age, gender, race/ethnicity, education, discipline or field of training, and identity (e.g., as a person living with a disability; as a person in recovery; as a person with lived experience, advocate, peer specialist, service provider, child, caregiver, partner or family member of a person with a disability) with multiple responses allowed. Additional items captured the participant’s background in behavioral health, disability or human services (e.g., number of years, type of agency or organizational affiliation, job title or role).

A semi-structured interview protocol allowed for open discussion and shifts in topics as initiated by the participant. The interview protocol included prompts pertaining to peer supports at the intersection of disability and opioid (mis)use: the definition of peer; the salience of disability in the opioid/substance use “world” and vice versa; the identification of models of peer support for this population; the essential elements of these models; barriers and facilitators of peer support at the disability/opioid (mis)use intersection; and recommendations for further development, adaptation and implementation of peer support models, improving care for this population in general. Research team members met regularly to debrief following interviews, compare notes regarding participants’ responses, and offer suggestions for the inclusion of participants and prompts in subsequent interviews.

### 2.3. Analysis

Background survey data were summarized to describe the sample. Interview recordings were professionally transcribed, and transcripts were uploaded into Dedoose, a web-based, mixed methods data management and analysis software [41]. Interview transcripts were reviewed by research team members, first sitting together, as the team developed the initial codes reflecting content themes, drawing from the interview protocol framework. As codes were elaborated and defined, and shared understanding developed, team members moved to code independently, meeting together bi-weekly to clarify any questions and confirm code application, to reach agreement on coding over time. A pooled Cohen’s Kappa coefficient was calculated to assess inter-rater reliability (0.87). Once coding was completed, memos were generated to summarize themes relevant to the research question. Memos were shared, discussed and refined, as necessary, to compile and integrate study findings. Preliminary findings were reviewed with the Peer Support Work Group for additional member checking, and triangulation of the data and their interpretation.

## 3. Results

### 3.1. The Participants

These were individuals with extensive professional and personal experience in human services, disability arenas, and in recovery themselves, who are very knowledgeable about peer supports. The vast majority (83%) of participants identified as having lived experience with the study topic. Over half (57%) identified as having a disability and nearly two-thirds (63%) identified as being in SUD recovery. More than half (58%) of the participants also reported having experience with a parent and nearly two-thirds (62%) with a family member with disabilities or in recovery. Over half identified as being peer specialists or recovery coaches (55%). Approximately one-quarter of the sample had educational backgrounds in each of these fields: social work, counseling, psychology, and education. Over three-quarters (79%) had a master’s degree or higher. All participants reported experience in the behavioral health, disability, or general human services fields, with more than half (57%) working 12 or more years in the field. Almost three-quarters of participants were female (71%), just under half (43%) were age 55 or older, and most identified as Caucasian or white (85%).

### 3.2. Essential Considerations

Several themes emerged to inform the adaptation and implementation of peer supports for individuals with disabilities and OUD/SUD. Participants described barriers to current supports, as well as recommended strategies for further consideration and development. Study participants, purposefully selected for their personal and professional experiences, all stressed the potential value of peer models and the need for careful consideration of issues to support access, engagement, participation, and benefit for both the peer specialist and the person with a disability being served.

#### 3.2.1. Accessibility and Model Fit

Study participants reflected on issues of service accessibility, though many of these challenges are not specific to the intersection of disability and opioid (mis)use. Accessibility was discussed in terms of the availability of relevant peer supports, the ability and willingness to access services, and the capacity to take advantage of relevant resources and materials. Most stakeholders found it difficult to identify any available programs or formal model of peer support that adequately met the needs of their clients. As one remarked, “we don’t really have models for this kind of care, it’s more a thought experiment”. In response, some established the type of services they recalled needing or wanting for others seeking OUD/SUD treatment. One participant adapted a 12-step program to meet a particular need of his clientele. Another opted to provide informal peer support with the goal of meeting people with disabilities where they congregated; he felt it was a “system that felt sorry for them because of their disability” rather than notice their need for sobriety. As this participant explained, “…the treatment system prioritized his disability over his addiction. Their attitude was …don’t worry about it. Being in a wheelchair…you guys are exempt from life; you don’t have to worry about this stuff”.

Physical accessibility is a common challenge among people with sensorimotor/mobility disabilities. Many traditional peer support meetings for people in SUD recovery (e.g., Alcoholics Anonymous) are held in church basements or other physically challenging spaces. As one participant pointed out, some meeting spaces (e.g., hotels or schools) may be more accessible and easier to navigate for people with disabilities than others. Another participant suggested that the space had to be accessible, but also “warm and welcoming”. They suggested that peer support models (e.g., Alcoholics Anonymous, Narcotics Anonymous) provide guidelines regarding meeting spaces, to ensure accessibility for disabled persons. In addition, location is an important issue for those with no access to transportation or who cannot drive. This individual lived in a rural area with limited public transportation, which he described as “…a huge factor in whether or not I can attend any type of meetings”.

Many participants remarked on the difficulties in finding OUD/SUD treatment in an accessible facility in general. Some participants suggested that legal rather than programmatic efforts were required to ensure accessibility, and the challenge to the system had to come from the very people cut off from treatment. As one participant acknowledged, “…if there’s nobody from the disability community bringing litigation against the alcohol and drug service system for its inaccessibility, ain’t much gonna’ happen”.

Participants reported that location matters in promoting participation in recovery services for individuals with disabilities. A participant recommended offering peer supports for substance use where people with disabilities “already are…having meetings in spaces that are frequently used by people with disabilities…is a way to expand opportunities for people with disabilities to get into recovery activities”. Another participant recommended the benefits of the co-location of services and supports: “…you need strong recovery support with…a hospital that specializes in your rehab services, at least close enough together so that someone who is disabled has access to get there and then to get back to the recovery center, especially if it is an inpatient recovery center”. Alternatively, the choice of location may keep people from participating, “…if…they’re trying to hide from certain people in their community”. The “right” location can be destigmatizing (e.g., a public library perhaps, rather than a treatment setting or clinic) and allow for participants to maintain confidentiality. Participants discussed dealing with the “double” stigma of both disability and OUD/SUD.

Study participants recommended the use of virtual/online video technology to minimize or overcome access barriers. Virtual meetings or supportive online communities may convey benefits particularly to individuals who avoid or are made uncomfortable by in-person situations. As one participant pointed out, “…a lot of neurodivergent people have intense social anxiety and could benefit from online meetings as well”. Through the use of web conferencing platforms (e.g., Zoom, Teams), individuals can choose to be seen, to share a photo, to block off a visual image completely or use the chat function. Individuals with disabling chronic health conditions that undermine in-person participation (e.g., mobility issues) may benefit from access to online support. Video and mobile technology (e.g., voice to text) can be further aids to communication, enabling text communications between persons and peers.

Another participant suggested the importance of having materials and resources in diverse formats (e.g., text at an appropriate reading level, video as well as text) addressing issues of sensory, cognitive, information processing, or attentional impairments, and language or health literacy challenges. This participant described an illustrated AA pamphlet that is in picture form. According to this participant, AA materials increasingly are becoming available in diverse formats, to “expand the reach [to] more of those types of communities that really can’t access, whether its sight, vision, cognitive things getting in the way…”.

Adaptation of the peer support model may be required to promote accessibility, acceptability, and use, particularly given the heterogeneous characteristics, experiences, and needs of the target population. Peer supports are not a “one size fits all” situation. As one participant recommended, “I think maybe there should be some kind of assessment in place first to make sure that it would be a fit for them”. Individuals with disabilities may be anxious, for example, in situations in which they feel they have no control (e.g., autistic people with exposure anxiety regarding sensory input they cannot control). Personal care attendants may not be allowed or made welcome to assist the disabled individual. Models relying on group interaction and sharing may be confusing or uncomfortable for individuals with cognitive or interpersonal challenges. The feeling of lack of control or discomfort and accompanying anxiety may even be more disabling. Peer support that fits well with a person’s disability and coping style may simply not be available.

#### 3.2.2. “Peerness” and Peer Match

The notion of “peerness” implies something about a person’s characteristics, conditions or life experiences, as well as their capacity to relate to the characteristics, conditions and life experiences of another. A peer was defined by study participants as “someone in recovery or someone with a similar disability” or both. For example, a peer might be someone who has incurred a brain injury consequent to a drug overdose or who suffered a brain injury and then “gotten hooked” on pain medication. A key characteristic is having lived experience and, preferably, similar lived experiences, so the peer understands the “challenges and frustration and a process people go through”. Aspects or characteristics of what is shared may vary in importance, as defined by the person receiving services, and may change over time. As one participant in a wheelchair contributed, “I was the only guy in a wheelchair going to 12-step meetings in my area at the time. However, they were able to identify with the emotions of addiction and recovery, and that’s when I felt like I belonged”.

People should be matched with a particular peer specialist not only on the basis of diagnosis or functioning, but on identity, that is, how they define and see themselves, according to participants. Only the person can rank the many components of their identity in terms of importance or meaning; meeting someone with all the same identities as their own was described as rare. Several participants recommended an initial assessment be conducted, to provide for the best match possible. Some participants suggested that a person may actually want somebody different from themselves to provide support and an alternative role model. Individuals with disabilities may have experiences of having been assigned labels or to groups or having been compared with people presumed to be peers, rather than being allowed to build identities or relationships of their own choosing. A person may attach shame to their status or situation (e.g., a “drug user”). Consequently, they may find a person with characteristics different from their own a better match. Ultimately, only the person served, together with the peer specialist, can evaluate the appropriateness of the match.

Participants agreed that peer specialists should be at a later stage in their own recovery, with experience coping with their conditions and navigating systems. The perfect peer may be hard to find, though abstractly, “would be somebody who’s in recovery from an opioid use disorder, who also had a disability challenge or disabling condition that they have dealt with…” This participant acknowledged that it is more likely to find a peer in recovery from opioid (mis)use, who can be trained in accommodations for people with disabilities, than the other way around.

Participants described what a peer offers and the ways in which they interact as important aspects of “peerness”. A peer who has had similar experiences and who has made progress in coping and recovery can serve as a positive role model and inspire motivation to make changes. Feeling understood is meaningful. As one participant described, “that…shows the power of at least understanding what it’s like to love someone or be someone with these multiple challenges and in recovery”. Being able to speak the same language, vernacular or jargon helps to build trust in the peer support relationship. Knowing that the peer has been through the same experiences reduces the shame associated with these experiences. “You know what I’m talking about. You know what addiction feels like. You know how hard this is”. The notion that “we’re in this together” introduces a mutuality that reduces stigma. “The more you can relate to somebody …or the more similarities that you see in them, the more likely the person is going to look to them as an example or as someone they can talk to and they can relate to…”

Perhaps one of the most powerful opportunities provided by peer support is the offering of hope that change is possible. The peer is “someone who could have the language and the hope piece”. One study participant described the value in knowing someone who is “like me, can be here with me, and offer me…living proof that recovery from what I’m going through is possible. ‘I’ve had this experience, let me help you through this. I know you’re in pain. I know you’re hurting, but it’s gonna’ be okay because I already did it and I came out of it…’ And once you’ve been through it and feel better about it, you want to share it”.

#### 3.2.3. Recruitment, Training, and Support

People with disabilities and/or opioid (mis)use may be recruited into peer support roles, motivated by the idea of helping others. Providing peer support gives a person a sense of purpose. One participant explained that, by recruiting people to provide peer support, you may give them a reason to stay clean and sober—to feel like they are “giving back”. “That’s a win-win-win-win for everybody, and I think that is where you have to figure out how can I best recruit those people, in a sense—have them ready, have them on standby, and let them know that what you’re doing has a great purpose. You’re doing this because you’re proving why this works”.

Peer specialists or recovery coaches in the mental health or substance use sectors may require training, and benefit from ongoing supervision and support to work with people with disabilities different from their own. First, study participants noted that peer specialists and recovery coaches need to question their assumptions and recognize what they do not know about people with other types of disabilities. Peers must ask the right questions to understand accommodations a person may require. Participants recommended a focus on functioning, rather than disability or diagnosis per se, to tailor or customize accommodations. As one participant suggested, “To boil it down to a reasonable chunk that’s digestible would be to talk about…things that people need to be successful in a program.” Disabilities may be “hidden”. A person with a hearing difficulty may need people to talk louder, or to meet in a space with minimal background noise; a person with a cognitive impairment may need to process information in smaller pieces or at a slower pace. Peer training should focus on how to question assumptions; how to be respectful of, sensitive to, and make practical adaptations for physical, cognitive and sensory needs and challenges; and on how to inquire about needs that may not be readily observable.

The behavioral health peer specialist’s feelings about working closely with someone with a disability should be explored, particularly if their own lived experience is dissimilar. One participant suggested a peer trained in the behavioral health perspective, but not knowledgeable about other disabilities, might feel like, “I don’t know how to do this. I’m scared. I don’t want to hurt this person”. Study participants suggested that training may help the peer be more patient, accepting, and empathetic, as well as provide practical tips. Peers must be “able to really sit with whatever is going on for the other person rather than trying to fix it right off the bat” and “use their experience in a way that doesn’t assume that the other person’s experience is exactly the same…”. The importance of ongoing supervision and support for a peer was underscored. Another participant recommended, “that [the peer] be trained also to understand that the person with the disability is really the expert on what they need”.

Concern was expressed by participants that a peer specialist or recovery coach coming from the behavioral health context may not understand or may misinterpret the actions of a person with other disabilities, for example, an autistic person’s perseverative behavior. Cross-training efforts have been successful in addressing these issues. For example, behavioral health peer specialists with certification have been provided additional training on key aspects of life with a long-term disability, “including education around some of the more common diseases and disorders that kind of lead to disability. But also, how to navigate the managed care system, what the disability system is. So, we want our certified peer specialists to…really be specialized”. In another program described by a study participant, developmental disability workers received training in substance use disorders, “to be able to understand the problems of the clients they’ve been struggling with, because it’s an alcohol or drug problem and they don’t know what to do with it, because they live in the developmental disability world”.

One participant underscored the likelihood that “everyone has trauma” and that “any kind of disability involves trauma—social, familial trauma, medical trauma”. Peers, who may themselves have experienced trauma, may benefit from supervision that takes their lived experiences into consideration, as they work together with and provide support to others with similar backgrounds and experiences. The best-case scenario, according to participants, is when peers are healthy and actively managing their own wellness and have had effective training and supervision “to manage their boundaries well”.

Finally, peer specialist training, as well as peer support services, must be readily accessible, physically as well as in terms of the format and usability of training resources and materials. As one participant pointed out, “…it’s not always that easy or affordable for people to get certified”. Individuals with physical disabilities may face additional accessibility barriers to training. For example, a person in a wheelchair with a personal attendant may not be able to participate in a training that does not take the personal attendant’s schedule into account. This participant suggested that to be more accessible, certified peer specialist training must be disability informed and accommodate different needs (i.e., not just the disability per se, but life circumstances accompanying the disability).

## 4. Discussion

Peer supports may be beneficial to individuals with disabilities with co-occurring OUD/SUD given the potential for overcoming barriers related to treatment seeking and engagement, and the evidence of benefits in target populations with similar needs [9,10,11,12,13,14,15,16,17]. Much of what emerged in our interviews with stakeholders regarding considerations in establishing models of peer support for individuals with disabilities and co-occurring OUD/SUD corroborates the published literature in behavioral health on treatment barriers [1,6,25,26,27,28] and potential benefits [9,10,11,12,13,14,15,16,17]. The translation of behavioral health peer support models would seem justifiable, particularly given the high prevalence of co-occurring behavioral health conditions for individuals living with these disabilities [4,42,43,44,45]. However, efforts to specify and test peer support models have largely occurred in the behavioral health domains [12]. To our knowledge, lessons learned in mental health and SUD domains have not been adapted nor informed the development and testing of peer support in other areas of disability (e.g., physical or sensorimotor disability, autism, or TBI). Adjustments or accommodations may be required, depending on the characteristics, preferences, and needs of the heterogeneous target population of individuals with disabilities and OUD/SUD, to promote accessibility, acceptability, and use of peer supports. Our findings add to an understanding of “peerness” and peer support that goes beyond the unidimensional definitions common in the extant literature described earlier (e.g., a peer is defined solely by lived experience with SUD). A more specialized approach or cross-training of peer specialists or recovery coaches could be helpful to tune into the unique needs of the person served and the ways in which their disability may impact their recovery journey.

Any model of peer support for individuals with disabilities requires particular and deliberate attention to the needs of a diverse and varied population. This is not unique to disability, as noted for people with mental illness or in the context of culturally sensitive care [46]. While issues of access to care must be addressed with any individual seeking OUD/SUD treatment [23,24], for individuals with disabilities, accessibility, in the broadest sense of the word, touches upon all aspects of care. It determines not only the ability and willingness to avail oneself of services, but the quality of said care and, in turn, the likelihood of engagement and retention in care. Issues of accessibility must be addressed at every level, from location to physical access to communication approaches and the formatting and presentation of relevant tools and resources. This is a hallmark of flexible, person-centered care, which is how peer services are often described, yet peers cannot do this all on their own. Such shifts in approaches to care may require systemic support, in terms of policy and practice guidelines, and payment models.

The co-location of OUD/SUD services, including peer supports, with disability/rehabilitation services may be the most effective strategy, where possible. A building “walk-through” at the OUD/SUD site would allow for the assessment of and attention to physical barriers that would make participation difficult. Alternate locations that are already accessible may be a better solution. Software packages facilitate the assessment of reading level; many packages for formatting images or creating simple videos are available. Having materials available in multiple formats will promote accessibility and use. Technology-based solutions, in general, may be useful in providing peer support to individuals with disabilities and OUD/SUD. Given the innovation in technology-based approaches to treatment and service provision during the COVID-19 pandemic, precedent has been set that supports continued use of virtual approaches in the future, to overcome a variety of challenges including transportation, mobility, language, scheduling of sessions, and payment.

Ample discussion exists within the peer support literature about the meaning and nature of peer support, “peerness”, and the match of the peer with individual [12,47]. How closely must the peer’s lived experience match or map onto that of the individual they seek to support? While any given response to this line of inquiry may vary depending on the context, nature, and setting of the work, existing models of peer support within the substance use recovery movement suggest that lived experience of addiction is essential to the work of a recovery coach or peer. Our stakeholders reiterated this point. It stands to reason then, that we should work to establish a preference in models of peer support for persons with disabilities that require one seeking to support another’s recovery to share a similar range of lived experiences, including that of a disabling condition, although this may not always be possible.

Equally important, the focus of peer support on coping with disability and/or dealing with OUD/SUD recovery may shift over time, depending on the current situation, status and needs of the person served. Take the case of a person recently injured in a serious car accident while driving under the influence of drugs, an example suggested by one of the study participants. This person’s immediate focus may be on learning to live with a new disabling condition (e.g., a TBI or the loss of mobility). However, the OUD/SUD contributing to the accident may also need to be addressed, but perhaps after a period of adjustment to the disability. The person may be better served by a peer with lived experience of disability first and then, when the timing is right, shift the focus to OUD/SUD recovery. Ideally, peer support could be available to address both concerns, but may not be available to be provided by the same peer specialist or recovery coach. This case highlights the importance of cross-training of peer specialists in diverse contexts, in hospital trauma units for example, to provide support to persons with complex physical and behavioral health needs.

Peer specialists may come to this work having trained in the behavioral health or disability service sectors. Additional training, perhaps offered through continuing education programs, should focus on the intersection of disability and OUD/SUD, and fill in the gaps for peer specialists originally trained in one sector or the other. While it may not be necessary or possible for a peer specialist to become an expert in both diverse disabilities and OUD/SUD, training must focus on which questions to ask, and the best ways to ask them, to promote an understanding of a person’s needs and to clarify any assumptions a peer specialist might make about a person served. Training and ongoing guidance should be provided to assist peer specialists in reflecting on their own assumptions and experiences, to recognize the ways in which their personal and professional experiences can inform or impede their work, to enhance the likelihood of benefit to the person served as well as the peer.

Peers are often seen as more authentic, less prescriptive, and less stigmatizing than practitioners providing more traditional behavioral health treatment [11]. Peers may possess a greater ability to meet the client “where they’re at” with empathy and without judgment, given their own experiences, to help guide the recovery process rather than advise or direct a particular course of treatment. Study participants, however, described challenges and offered recommendations for meeting the unique, diverse needs of individuals living with both disability and OUD/SUD. These challenges reflect the broader literature about health care professionals addressing disability, such as a lack of knowledge about or discomfort with disability and accommodations [48,49]. While a “one size fits all” approach may not be possible or advisable, certain considerations would seem to be cross-cutting.

### Study Limitations

This study was exploratory. The data were qualitative, obtained from a purposefully selected sample of known experts, with both extensive professional and personal experience in disability and OUD/SUD treatment and recovery. While the sample is small, participants reflected diverse training, disciplines, and roles, and drew from years of experience in the field. The majority identified as having a disability and as being in SUD recovery. Many had experience providing services, including peer supports, as the intersection of disability and OUD/SUD. A number had, themselves, created innovative solutions, building existing models or services to meet the needs of individuals at the intersection. In-depth interviews with these experts were a warranted first step in informing the development of relevant, accessible, potentially effective peer supports.

## 5. Conclusions

Clearly, peer supports at the intersection of disability and OUD/SUD hold the promise of benefit for both the individual person served, as well as the peer providing the service. Better methods are essential to identifying and addressing the oftentimes complex characteristics, daily lives, and multifaceted needs of individuals with co-occurring disability and OUD/SUD to overcome barriers to services, facilitate engagement, and promote recovery. Peers may bring insights in addressing these challenges and, with relevant training and support, contribute to improved outcomes for individuals with disabilities and OUD/SUD. Future research should include larger samples of participants receiving and/or not receiving services, to obtain data regarding their perceptions of needs, barriers, the elements of effective peer support, and their perspectives on the definition of “success” in living with disability and OUD/SUD. Recommendations for peer support models, to be developed and tested, will likely vary depending in part on the context in which they are provided. It will be important to begin to understand the contingencies operating in diverse contexts and tailor peer supports to both the individuals being served as well as the context in which they are identified, referred, and engaged in services. Current training and certification guidelines may require modification or enhancement to include training and support for peers working with individuals at the intersection and their complex needs and lives. Policy change may be required to ensure reimbursement for effective peer supports. Peer support efforts at the intersection of disability and OUD/SUD have a strong foundation upon which to build in the disability and behavioral health sectors. Research findings and practice knowledge from both sectors must be integrated to address the needs of individuals living with these complex challenges, to promote optimal functioning, recovery, and the achievement of life goals.

## Data Availability

Data are available from the authors upon request and with discretion, given the size and nature of the sample, to ensure confidentiality.

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
