# Peer review of "Peer Support at the Intersection of Disability and Opioid (Mis)Use: Key Stakeholders Provide Essential Considerations"

_ijerph, 2022, doi:10.3390/ijerph19159664_

Round 1

Reviewer 1 Report

This is an interesting study which may provide some guidance for stakeholders regarding the accessibility of peer support for opioid use disorder in individuals with disabilities in the US.

It should be specified by the authors that the disabilities mentioned in the study are completely heterogenous as is their association with opioid use. Patients with spinal cord or traumatic brain injury for example are highly likely to require opioid analgesics for pain control, individuals with mood disorders or psychosis may self-medicate with opioids either for the primary condition or to alleviate medication side effects. They are not significantly more likely to require opioids for analgesia though. Individuals with autism spectrum disorder may have difficulty accessing any illicit drugs, depending on the severity of their condition.

The sample characteristics should be included in the form of a table, most important in addition to age and sex are in this case is occupation, peer support group experience and groups of disabled individuals with which each participant has personal experience. 

Reviewer 2 Report

This is an excellent paper, on many different levels. Coming from the perspective of behavioral health, it brings the approach of intersectionality to the qualitative study of individuals with disability and opioid use disorder. The aim of the paper is to explore what models of peer support might be employed to help persons with disability recover from OUD. The paper is excellently written and excellently argued; it is well researched and highly original. It shows both how questions of peer support for persons at the intersection between disability and OUD is specific and how the model suggested in this paper may be applied to other contexts. What is particularly original is that the individuals recruited for the qualitative interviews possessed both personal and professional experiences with being/working at the intersection of disability and OUD. Another really strong point of the paper is that the voices of the study participants could be heard, and thus they were present in the study in a sense that epitomized the idea of working with rather than working about a given community. Another asset of the paper was its notion of “peerness”, i.e. that categories in which individuals may act as peers are not predefined, but are rather matched to the person who needs support. On both a theoretical and a practical level, this is really a strong paper. The only thing that might still be added, but this is entirely optional, is whether the authors might elaborate on behavioral health as a field, and potential collaborations with other fields and disciplines, such as sociology, but also and especially disability studies.

Reviewer 3 Report

Thank you very much for the opportunity to review the submitted manuscript. I did it with great pleasure, especially since my research interests are both addiction and disability. Interesting text, clearly described research process, interestingly presented results. In my opinion, however, some modifications and additions should be made in order for the text to meet the criteria of sound scientific work.

Section: Participants I must admit that I personally avoid presenting data as a percentage if the research sample consists of a dozen or so people, and besides, these are qualitative research. In my opinion, it looks a bit unnatural that "Most of them had a master's degree or higher (78.6%)." Most of the sixteen people - so 12? I think so. And "Most of the participants are women (71.4%)" - are 11 women? Admittedly, this is not a glaring mistake, but somehow it seems unnatural to me. Please see the sentence the authors wrote - after removing the percentages, it shows the same data, but without this strange math: "More than half of the participants said they had a disability and almost two-thirds said they were recovering from SUD." I think it is worth presenting it in a clearer and simpler way.

In my opinion, the Discussion chapter needs to be supplemented and rewritten. The written content is likely to be important, but it is more the end of the study than an actual discussion with other similar studies. In my opinion, part of the introduction and chapters 1.2 to 1.4 could be moved to the Discussion section, and the results of the publications cited there could be slightly expanded and simply discussed.

The research is interesting, but I think that the description (in the manuscript under review) lacks the complement of a discussion with other data. A significant part of the Discussion section could be transferred to the conclusion, in which there are conclusions. This is good text and should not be deleted.

Technical comments: please adapt the References section to the journal's guidelines.

Round 2

Reviewer 3 Report

I thank the authors for their forbearance and making corrections and additions. In my opinion, now it is a very good scientific article. 

Good luck in your future work!